# Design and Experimental Study of a Bi-Directional Rotating Stubble-Cutting No-Tillage Planter

**Huibin Zhu** [1], **Xian Wu** [1], **Cheng Qian** [1], **Lizhen Bai** [1,*], **Shiao Ma** [1], **Haoran Zhao** [1], **Xu Zhang** [1] and **Hui Li** [2]

1 Faculty of Modern Agricultural Engineering, Kunming University of Science and Technology, Kunming 650500, China
2 Shandong Academy of Agricultural Machinery Science, Ji'nan 250100, China
* Correspondence: lzhbai@kust.edu.cn; Tel.: +86-189-8748-6806

**Abstract:** In view of the problem of a large amount of stubble and straw in Southwest China, it is difficult to carry out no-tillage sowing operations. Based on the principle of supported cutting, a bi-directional rotating stubble-cutting no-tillage planter was designed. According to the extracted left and right mouth contours of Batocera horsfieldi (Hope), the blade curve of a bi-directional rotating cutterhead was designed. The discrete element models were established regarding 'bi-directional rotating disc cutter, straw and soil', 'fertilizer apparatus and fertilizer', and 'opener and soil' in the Extended-Domain-Eigenfunction Method (EDEM) software. The optimal working parameters were analyzed using a quadratic regression orthogonal rotation test and response surface methodology. Accordingly, the prototype was manufactured and the field performance test was carried out. The best working parameters of the machine were as follows: the forward speed of the machine was 0.9 m·s$^{-1}$, the cutter spacing was 60 mm, the forward speed was 150 r·min$^{-1}$, and the reverse speed was 313 r·min$^{-1}$. The field experiment results showed that the average cutting rate of corn straw was 95.72% using the anti-blocking device when the straw mulching amount was 1.63 kg·m$^{-2}$. The average sowing depth was 5.4 cm, the average fertilization depth was 10.1 cm, and the average seed–fertilizer spacing was 4.7 cm. The qualified rates of sowing depth, fertilization depth, and spacing were 88.89%, 100%, and 100%, respectively. The designed bi-directional rotating stubble-cutting no-tillage planter can meet the requirements of no-tillage sowing in Southwest China. This study can provide reference for the design and improvement of no-tillage planters under the conditions of a large amount of stubble and straw.

**Keywords:** anti-blocking; bionics; stubble cutting; EDEM simulation; no-tillage planter; design optimization

## 1. Introduction

The maize-producing area of Southwest China, as one of the main maize-producing areas in China, accounted for 11.1% of the total planting area in 2020 [1]. The traditional agricultural tillage mode, soil degradation, and soil erosion of cultivated land are aggravated, which seriously threatens food security and ecological health in Southwest China [2,3]. In order to effectively reduce soil erosion, straw burning, and greenhouse gas emissions, it is urgent to develop conservation tillage technology in the maize planting area of Southwest China.

During the no-tillage seeding operation, the straw covering on the ground will cause blockage and wind the opener, so in order to ensure the smooth no-tillage seeding it is necessary to install a stubble-cutting and anti-blocking device on the no-tillage planter to crush or clean the previous crop stubble and straw on the seeding belt. At present, the anti-blocking devices are divided into active ones [4] and passive ones [5]. The active device is driven by the tractor output shaft to conduct stubble-cutting and crushing operations [6] to achieve the purpose of anti-blocking, and the active straw cleaning type is used to

clean the straw on the seeding belt [7], such as stubble cleaning of the seeding belt [8], spring-tooth cleaning [9], etc. The passive type relies on its own gravity and the sliding cutting or chopping [10] of the stubble-cutting device for stubble-cutting and anti-blocking, such as the notched disc knife type [11], spiral cutting type [12], etc.

The climate of Southwest China belongs to the subtropical monsoon climate. There are many weeds in cultivated land and the coverage of straw and stubble is large after crop harvests. The corn harvest method in Southwest China is mainly manual, and corn straw is mostly not crushed. Instead, the whole straw falls into the ground. The active and passive anti-blocking devices cannot complete the stubble-cutting and anti-blocking operations well and cannot satisfy the passing of the machine either.

In view of the above problems, based on the principle of supported cutting, the equation fitting of the contour curve of the B. horsfieldi (Hope) mouthparts was carried out using the bionic principle, and the optimal cutting curve was designed. The simulation experiment of EDEM was established and a bi-directional rotating stubble cutting anti-blocking device was designed. The quadratic regression orthogonal rotation combination test was carried out to determine the optimal parameter combination of the anti-blocking device. The prototype was completed, and the operation performance of the device was tested in the field.

## 2. Materials and Methods

### 2.1. Mechanism and Working Principle of the Whole Machine

The bi-directional rotating stubble-cutting no-tillage planter can complete stubble cutting, fertilization, and sowing at one time. The planter mainly includes a roller, soil wheel, ground wheel, fertilization furrow opener, fertilizer apparatus, rotary disc cutter, reverse disc cutter, T series gearbox, depth limiting disc, and frame, as shown in Figure 1.

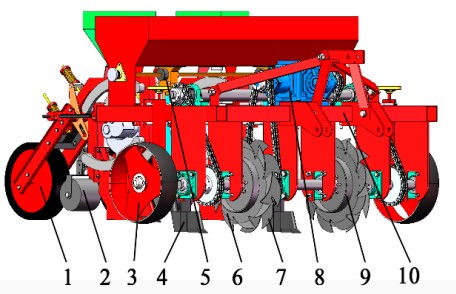 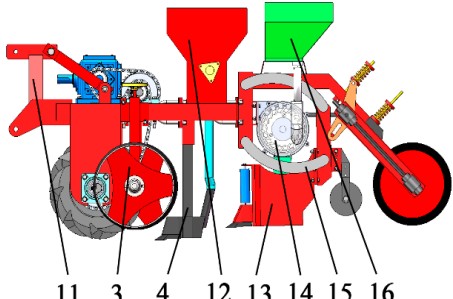

**Figure 1.** Structural diagram of the bi-directional rotating stubble-cutting no-tillage planter. Note: 1. Compression wheel; 2. Covering wheel; 3. Land wheel; 4. Fertilization furrow opener; 5. Exhaust fertilizer; 6. Forward disc cutter; 7. Reverse disc cutter; 8. T series gearbox; 9. Depth-control disc; 10. Frame; 11. Three-point suspension device; 12. Fertilizer box; 13. Sowing furrow opener; 14. Spoon wheel seed metering device; 15. Imitation device; 16. Seed box.

The bi-directional rotating stubble-cutting no-tillage planter belongs to the suspended planter. During operation, the power of the tractor power output shaft is transmitted to the forward disc cutter and the reverse disc cutter through the anti-blocking transmission system. The rotating forward disc cutter and the reverse disc cutter create a relatively supported cutting process to cut the straw. The depth-control disc on one side of the forward disc cutter and the reverse disc cutter controls the ditching depth of the disc cutter, and the straw on the seeding belt can be compressed to improve the cutting performance. The fertilizer furrow opener and sowing furrow opener located behind the disc cutter completes the fertilization and sowing operations, respectively, and the soil on the surface of the seed ditch is pressed by the compression wheel after soil erosion to prevent moisture loss.

## 2.2. Design and Analysis of Key Components

### 2.2.1. Design of the Bionic Disc Cutter

The adults of B. horsfieldi feed on the leaves of the trunk and have a strong mouth cutting ability [13]. Therefore, in order to seek the edge curve of the disc cutter, the contour of the mouthparts of the B. horsfieldi was extracted using the bionic principle. When the blade is cut by the mouthparts, the opposite movement of the left and right mouthparts forms a supporting cutting mechanism. By studying and analyzing the contour curve of the mouthparts, it can provide the basis for the blade design of the disc cutter.

The contour curve of an organism was obtained through image processing technology, and the flow chart of extraction of the contour curve of B. horsfieldi mouthparts is shown in Figure 2.

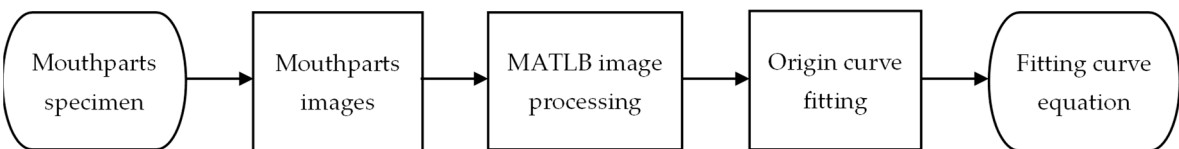

**Figure 2.** Process diagram of contour curve extraction.

The mouthparts of B. horsfieldi were observed with a microscope, and the images of mouthparts were extracted, as shown in Figure 3. The images were processed using MATLAB software [14], and the images were gray-scale processed to simplify the pixel matrix. After erosion and dilation of the gray-scale images, the interference points were removed and the boundaries of the images were clearer [15,16]. The grayscale image was binarized, and the filling function was used to fill the binary image. Finally, the Canny segmentation function was used to segment the edge of the image, and the contour curves of the mouthparts of the B. horsfieldi were obtained. The coordinates of the pixels of the contour curves were derived. The image processing flow chart is shown in Figure 4. The coordinates were imported into Origin to obtain the curve equation.

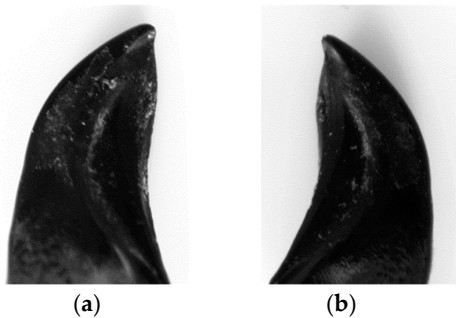

(**a**)                              (**b**)

**Figure 3.** Mouthparts of batocera horsfieldi. (**a**) Right mouthpart; (**b**) left mouthpart.

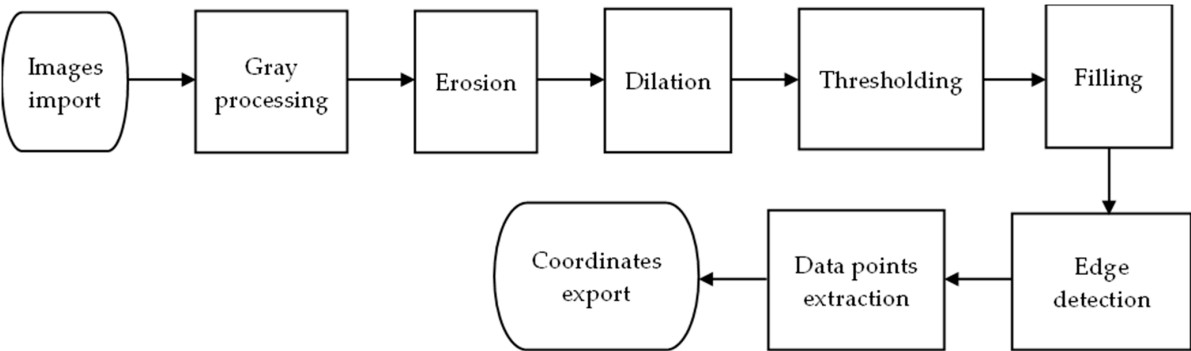

**Figure 4.** Process diagram of image processing.

The disc cutter radius should meet the requirement that the straw will not be moved and blocked along the ground after being cut off [17]. The larger the disc cutter diameter is, the better the cutting performance is, but the power and the torque on the cutter shaft will be larger at the same time [18]. The main stubble depth of maize in Southwest China was measured, and the main stubble depth was 50~90 mm. The disc cutter soil depth was selected as 100 mm in order to prevent the opener from blocking. The radius of the cutter shaft and the distance between the cutter shaft and ground were determined to be 20 and 120 mm, respectively, to ensure the strength of the cutter shaft and prevent interference, thus, the radius of disc cutter is 240 mm. The fitting curve was drawn using CAD software and the scaling was adjusted. The equations of the inner edge $L_1$ and the outer edge $L_2$ of the right disc cutter and the inner edge $L_3$ and the outer edge $L_4$ of the left disc cutter were obtained as shown in Formula (1), and the curves of each blade are shown in Figure 5.

$$\begin{cases} L_1 = -76.22 - 0.32x + 0.0024x^2 \\ L_2 = 9.04 - 1.63x + 0.00435x^2 \\ L_3 = -73.66 + 0.30x - 0.00475x^2 \\ L_4 = -122.70 + 1.60x - 0.00609x^2 \end{cases} \tag{1}$$

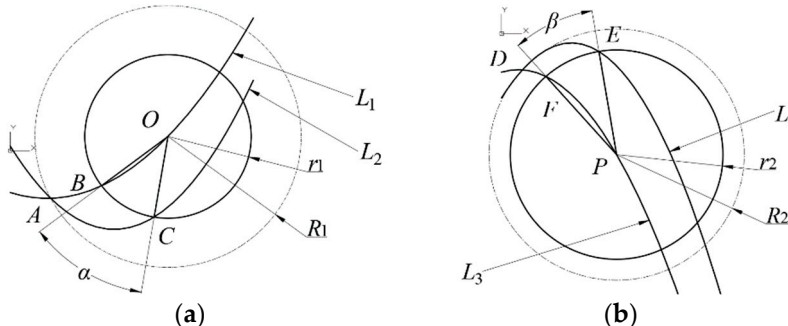

(a)  (b)

**Figure 5.** Edge curve of the disc cutter. (**a**) Right disc blade curve; (**b**) left disc blade curve. Note: points O and P are the centers of the right disc cutter and left disc cutter, respectively; points $\alpha$ and $\beta$ are the radian angles of the right disc cutter and left disc cutter, respectively, (°); point D is the intersection point of the inner blade and outer blade of the left disc cutter; point F is the intersection point between the inner blade of the left disc cutter and the tooth root circle; point E is the intersection point between the outer blade of the left disc cutter and the tooth root circle.

Point O is the circle center of the right disc cutter and its coordinate is $(x_1, y_1)$. Then, the equations of the tooth top circle $R_1$ and the tooth root circle $r_1$ of the right disc cutter teeth were obtained and shown in the Equations (2) and (3), respectively:

$$(x - x_1)^2 + (y - y_1)^2 = R_1^2 \tag{2}$$

$$(x - x_1)^2 + (y - y_1)^2 = r_1^2 \tag{3}$$

The functional relationship between the number of cutter teeth $N_1$ and the arc angle of cutter teeth of the right disc cutter $\alpha$ is shown in Equation (4).

$$\alpha = \frac{360°}{N_1} \tag{4}$$

Formula (5) can be obtained according to the cosine theorem:

$$\cos \angle BOC = \cos \alpha \tag{5}$$

Points B and C are the points on the inner blade curve $L_1$ and the outer blade curve $L_2$ of the right disc cutter, respectively, and point A is the intersection of $L_1$ and $L_2$. By

combining the above formula, the functional Formula (6) of the number of teeth $N_1$ and the tooth height $H_1$ of the right disc cutter can be obtained as follows:

$$H_1 = 258.09 - 29.61N_1 + 1.26N_1^2 - 0.018N_1^3 \ (4 \leq N_1 \leq 24), N_1 \in Z \tag{6}$$

Similarly, the function Formula (7) of the number of teeth $N_1$ and the tooth height $H_1$ of the left disc cutter can be obtained as follows:

$$H_2 = 129.54 + 17.68N_2 - 3.99N_2^2 + 0.15N_2^3 \ (4 \leq N_2 \leq 24), N_1 \in Z \tag{7}$$

The parameters of tooth number and tooth height of left and right disc cutters can be obtained using Formulas (6) and (7). In the operation process, the soil will produce large torque and wear on the disc cutter. Therefore, the tooth height does not easily become too large, 40~80 mm is more appropriate [19], and the number of teeth is 9~11.

### 2.2.2. Design of the Fertilizer Apparatus

The fertilizer apparatus is one of the key components of the no-till planter, which can be classified into the external groove wheel type, star wheel type, screw type, and horizontal scraper type [20,21]. The external grooved wheel fertilizer apparatus is mainly composed of a fertilizer box, an external grooved wheel, a fertilizer tongue, a fertilizer tongue shaft, a retaining ring, and a fixed plate [22], as shown in Figure 6. In order to improve the fertilizer discharge uniformity of the external grooved wheel fertilizer apparatus, three external grooved wheels with different notch shapes are designed, as shown in Figure 7. The amount of fertilizer discharged per revolution of the external grooved wheel fertilizer apparatus can be calculated according to Equation (8) [23].

$$\begin{cases} q_t = q_1 + q_2 \\ q_1 = \dfrac{\rho NSL\eta}{1000} \\ q_2 = \dfrac{2\pi RL\rho\gamma}{1000} \end{cases} \tag{8}$$

In the formula, $q_t$ is the fertilizer discharge per revolution of the fertilizer apparatus, g; $q_1$ is the quality of fertilizer discharged per rotational forcing layer, g; $q_2$ is the quality of fertilizer discharged per rotary drive layer, g; $\rho$ is the density of fertilizer, g·cm$^{-3}$; N is the number of grooves; S is the cross-sectional area of the groove, mm$^2$; L is the effective working length of the groove wheel, mm; $\eta$ is the filling coefficient of fertilizer in the groove; R is the radius of the groove wheel, mm; $\gamma$ is the driving coefficient of the driving layer fertilizer.

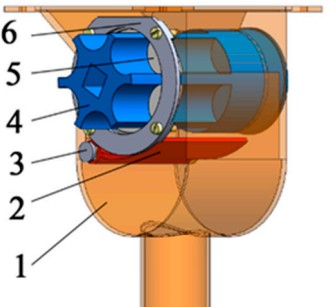

**Figure 6.** Structural diagram of the outer-groove wheel fertilizer. 1. Fertilizer box; 2. External groove wheel; 3. Fertilizer tongue; 4. Fertilizer tongue shaft; 5. Retaining ring; 6. Fixed plate.

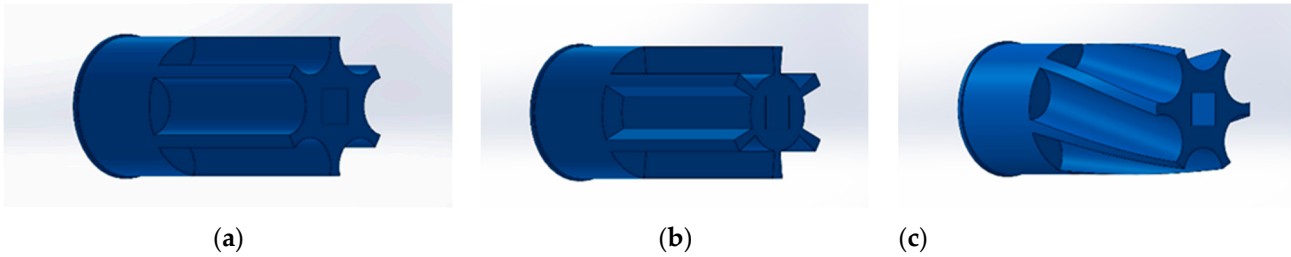

<div align="center">(<strong>a</strong>)   (<strong>b</strong>)   (<strong>c</strong>)</div>

**Figure 7.** Different types of outer-groove wheel. (**a**) Circular arc straight groove type; (**b**) ladder-arc straight groove type; (**c**) circular arc chute type.

### 2.2.3. Selection of Ditcher

The ditcher is an important component of a no-tillage planter, and its ditching quality directly affects the emergence rate of crops. It can be mainly classified into a double disc type, hoe type, and core type [24,25]. In order to improve the sowing quality, the optimal disc cutter was selected. The diameter of the double disc opener is 350 mm, the disc angle is 16°, and the gathering point angle is 15°. The opening angle of the core-share opener is 60°, the soil entry angle is 25°, and the width is 120 mm. The opening angle of the hoe opener is 22°, the soil entry angle is 30°, and the width is 40 mm.

### 2.3. Discrete Element Simulation Test of Key Components

The numerical simulation test method can greatly shorten the test cycle and cost and facilitate the in-depth analysis of the mechanism of action between 'machine, soil and plant'. Therefore, based on the discrete element method, the key component device was simulated to determine its structural parameters and motion parameters. In the simulation experiment, EDEM is the discrete element software, which has been widely used in the field of agricultural engineering in recent years.

### 2.3.1. Discrete Element Model and Experimental Design of the Anti-Blocking Device

The discrete element model of 'anti-blocking device, straw and soil' was established using EDEM software, and the material of anti-blocking device was 65 Mn. According to relevant literature [26,27], model parameters and contact parameters are shown in Tables 1 and 2.

**Table 1.** Intrinsic parameters of models.

| Materials | Poisson's Ratio | Density/(kg·m$^{-3}$) | Shear Modulus/Pa |
|---|---|---|---|
| Steel | 0.30 | 7800 | $8.0 \times 10^{10}$ |
| Straw | 0.40 | 470 | $1.7 \times 10^{6}$ |
| Soil | 0.38 | 2680 | $1.2 \times 10^{6}$ |

**Table 2.** Contact parameters of models.

| Contact Model | Static Friction Coefficient | Dynamic Friction Coefficient | Recovery Coefficient |
|---|---|---|---|
| Straw–straw | 0.14 | 0.08 | 0.49 |
| Soil–soil | 0.84 | 0.10 | 0.55 |
| Straw–soil | 0.50 | 0.01 | 0.50 |
| Steel–straw | 0.23 | 0.12 | 0.66 |
| Steel–soil | 0.60 | 0.10 | 0.30 |

The soil adopted Hertz Mindlin with the JKR model, with a particle radius of 4 mm, contact radius of 4.5 mm, and surface energy of 12.73 j·m$^{-2}$. The Hertz Mindlin with bonding model was adopted for corn straw. The contact radius of corn straw was 1.2 mm,

the normal stiffness per unit area was $9.361 \times 10^7$ n·m$^{-3}$, the shear stiffness per unit area was $9.845 \times 10^7$ n· m$^{-3}$, the critical normal stress was $1 \times 10^8$ Pa, the critical shear stress was $1 \times 10^8$ Pa, and the bond key radius was 2 mm [28]. The soil tank size was $1000 \times 400 \times 300$ mm, and soil particle diameter was 4 mm. In total, 24 straws with a diameter of 30 mm and a length of 300 mm were taken. The fertilization depth of no-tillage sowing was 80~100 mm [29], so the operation depth of the anti-blocking device was 100 mm. The discrete element model is shown in Figure 8.

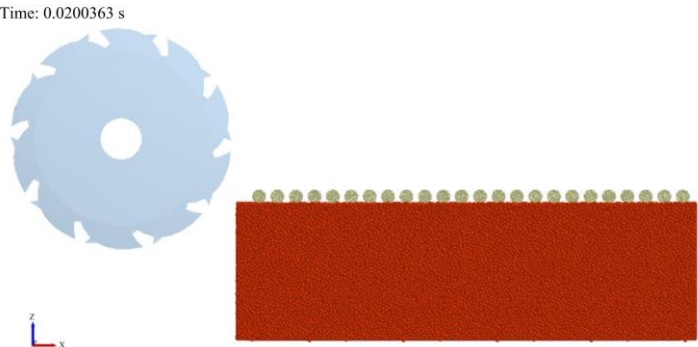

**Figure 8.** Discrete element model of the anti-blocking device.

According to the previous theoretical analysis, the operation performance of the anti-blocking device was affected by the forward speed of the implement, the cutter head spacing, the forward rotating cutter head speed, and the reverse rotating cutter head speed [30]. Therefore, the type of the forward rotating disc cutter, the number of teeth of the disc cutter, the forward speed of the implement, the cutter spacing, the forward rotating speed of the disc cutter, and the reverse rotating speed of the disc cutter were selected as the test factors of the simulation test. In combination with the ditching width of the ditcher, the cutter head spacing was 20~60 mm. According to the national standard of the rotary cultivator [31], the forward speed range of the implement was 0.56~1.39 m·s$^{-1}$, and the rotating speed range of the bi-directional rotating cutter heads was 150~350 r·min$^{-1}$.

The straw crushing degree and power were taken as the test indexes. When the positive and negative disc knives cut straw, it would be broken and cracked. The breaking degree of bonding in the simulation test replaced the breaking degree of straw. The operating power was calculated according to Equation (9) [32].

$$P = \frac{M_z n_z + M_f n_f}{9550} + \frac{v_m (F_z + F_f)}{1000} \tag{9}$$

In the formula, P is the operating power, Kw; $M_z$ is the torque on the rotary cutter, N·m; $M_f$ is the torque to reverse the cutterhead, N·m; $n_z$ is positive rotating speed, r/min; $n_f$ is counter rotating speed, r· min$^{-1}$; $v_m$ is forward speed, m·s$^{-1}$; $F_z$ is the forward resistance during operation, N; $F_f$ is the forward resistance during operation, N.

The types of forward and reverse rotary disc cutters were selected as the right curve disc cutter and left curve disc cutter, respectively [30]. Additionally, the number of disc cutter teeth was 9 [30]. The quadratic regression rotation orthogonal combination test with four factors and five levels was carried out. The test factors and levels are shown in Table 3. The test indexes were the bond fracture number $Y_1$ and operating power $Y_2$.

**Table 3.** Experimental factors and levels.

| Level | Factor | | | |
|---|---|---|---|---|
| | Machine Forward Speed $X_1$/(m·s$^{-1}$) | Disc Spacing $X_2$/(mm) | Forward-Rotation Speed $X_3$/(r·min$^{-1}$) | Reverse-Rotation Speed $X_4$/(r·min$^{-1}$) |
| 1.682 | 1.67 | 73.64 | 81.82 | 81.82 |
| 1 | 1.39 | 60 | 350 | 350 |
| 0 | 0.975 | 40 | 250 | 250 |
| −1 | 0.56 | 20 | 150 | 150 |
| −1.682 | 0.28 | 6.36 | 418.18 | 418.18 |

2.3.2. Discrete Element Model of the Fertilizer Discharge Device

In order to study the influence of different notch shapes on the fertilizer discharge uniformity of the external grooved wheel fertilizer apparatus, the discrete element model of "external grooved wheel fertilizer apparatus and fertilizer" was established based on the discrete element method, as shown in Figure 9, and the simulation test was carried out. The intrinsic parameters and contact parameters of fertilizer particles and external trough wheel fertilizer apparatus are shown in Table 4, and the particle radius was 1.8 mm [33]. In order to evaluate the effect of different notch shapes on the uniformity of fertilizer discharge of external grooved wheel fertilizer apparatus, the simulation test was carried out with the coefficient of variation of fertilizer discharge uniformity as the test index. The variation coefficient of the fertilizer discharge uniformity can be calculated according to Equations (10)–(12) [34,35].

$$M = \frac{1}{n} \sum_{i=1}^{n} m_i \tag{10}$$

$$S = \sqrt{\frac{\sum_{i=1}^{n} (m_i - M)}{n - 1}} \tag{11}$$

$$\delta = \frac{S}{M} \times 100\% \tag{12}$$

In the formula, $m_i$ is the mass of fertilizer in the ith plot, g; n is the number of measurement cells; M is the average mass of fertilizer in each plot, g; S is the standard deviation of fertilizer quality in each plot; δ is the coefficient of variation of fertilizer uniformity.

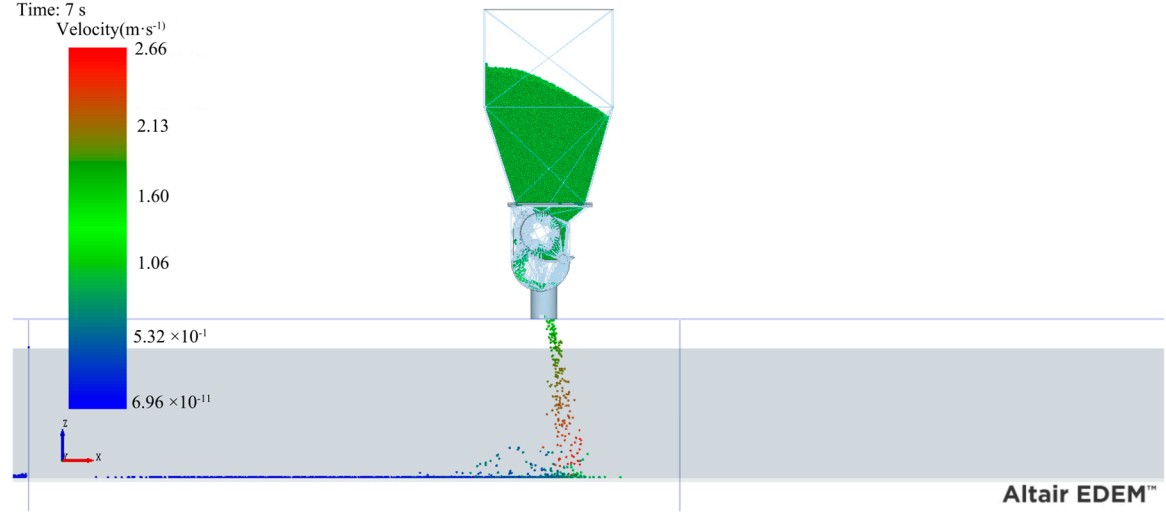

**Figure 9.** Discrete element model of fertilizer apparatus.

**Table 4.** Intrinsic parameters and contact parameters.

| Materials | Eigen Parameters | Value | Materials | Contact Parameters | Value |
|---|---|---|---|---|---|
| Fertilizer particles | Poisson's ratio | 0.25 | Fertilizer and fertilizer | Recovery coefficient | 0.11 |
| | Shear modulus/Pa | $2.8 \times 10^7$ | | Static friction coefficient | 0.30 |
| | Density/(kg·m$^{-3}$) | 1320 | | Dynamic friction coefficient | 0.10 |
| Fertilizer exhauster | Poisson's ratio | 0.43 | Fertilizer apparatus and fertilizer | Recovery coefficient | 0.41 |
| | Shear modulus/Pa | $1.3 \times 10^9$ | | Static friction coefficient | 0.32 |
| | Density/(kg·m$^{-3}$) | 1240 | | Dynamic friction coefficient | 0.18 |

2.3.3. Discrete Element Model of the Opener Device

The ditcher should meet the technical requirements of conservation tillage with low soil disturbance, and the resistance of the ditcher should not be too large to reduce the forward resistance of the whole machine. The resistance and shape of the ditcher were selected as the evaluation basis, and the simulation test of the ditcher was carried out based on the discrete element method. The model is shown in Figure 10.

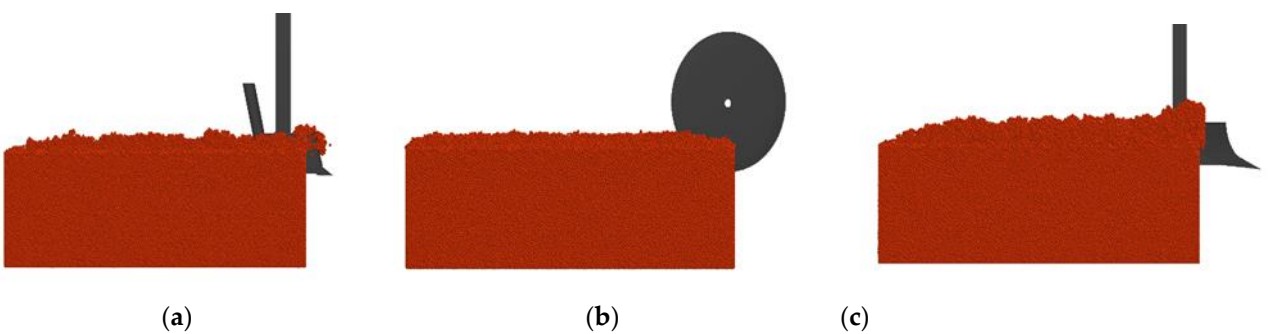

(**a**)　　　　　　　　　　　　　(**b**)　　　　　(**c**)

**Figure 10.** Discrete element model of the opener. (**a**) Hoe openers; (**b**) double disc openers; (**c**) core-share openers.

**3. Results and Discussion**

*3.1. Experimental Results and Surface Response Analysis of the Anti-Blocking Device*

The test results were analyzed using Design-Expert, and the insignificant factors from the regression analysis were added to the residual term analysis. The variance analyses of the number of broken bonds and operating power are shown in Tables 5 and 6. The regression equations $Y_1$ and $Y_2$ were obtained. The regression models were very significant ($p < 0.01$).

$$Y_1 = 132{,}994.00 - 4434.55X_1 - 10{,}436.50X_2 + 11{,}022.8X_4 + 10{,}546.3X_1X_2 + 11{,}555.00X_1X_4 + 12{,}468.90X_2X_4 - 6931.59X_1^2 - 2370.75X_4^2 \tag{13}$$

$$Y_2 = 5.95 - 0.69X_1 + 0.72X_2 + 1.21X_3 + 0.94X_4 + 0.59X_1X_4 - 0.54X_2X_4 - 0.23X_3^2 + 0.18X_4^2 \tag{14}$$

The response surface of the interaction factors on the number of bond fractures $Y_1$ was obtained using 'Design Expert' software, as shown in Figure 8.

It can be seen from Figure 11a that when the cutter head spacing was at the initial stage, the number of broken bonds decreased with the increase in the forward speed of the machine. However, with the increase in the cutter head spacing, the influence of the increase in the tool forward speed on the number of broken bonds changed from a decreasing trend to an increasing trend. The constraint conditions are shown in Formula (15). The reason for this change is that when the spacing is the initial stage, the increase in the forward speed of the machine reduces the cutting frequency of the anti-blocking device to the straw, and the influence of the cutter head spacing on the number of broken bonds is greater than the

forward speed of the machine. With the increase in the cutter head spacing, the number of broken bonds increases with the increase in the forward speed of the machine. When the forward speed of the machine tool is the initial value, the number of broken bonds increases with the increase in cutter head spacing, and the number of broken bonds increases more obviously with the increase in the forward speed of the machine tool. The reason is that the increase in the cutter head spacing will tear the cutting surface of the straw, resulting in an increase in the number of broken bonds.

**Table 5.** Variance analysis of the number of broken bonds.

| Test Index | Source | Sum of Squares | Freedom | Mean Square | F Value | p Value |
|---|---|---|---|---|---|---|
| | Model | $4.56 \times 10^9$ | 8 | $5.70 \times 10^8$ | 29.94 | <0.0001 |
| | $X_1$ | $1.11 \times 10^8$ | 1 | $1.11 \times 10^8$ | 5.84 | 0.0299 |
| | $X_2$ | $6.16 \times 10^8$ | 1 | $6.16 \times 10^8$ | 32.35 | <0.0001 |
| | $X_4$ | $6.87 \times 10^8$ | 1 | $6.87 \times 10^8$ | 36.08 | <0.0001 |
| Number of broken bonds | $X_1X_2$ | $3.69 \times 10^8$ | 1 | $3.69 \times 10^8$ | 19.35 | 0.0006 |
| | $X_1X_4$ | $4.42 \times 10^8$ | 1 | $4.42 \times 10^8$ | 23.23 | 0.0003 |
| | $X_2X_4$ | $5.15 \times 10^8$ | 1 | $5.15 \times 10^8$ | 27.05 | 0.0001 |
| | $X_1^2$ | $7.64 \times 10^8$ | 1 | $7.64 \times 10^8$ | 40.08 | <0.0001 |
| | $X_4^2$ | $8.93 \times 10^7$ | 1 | $8.93 \times 10^7$ | 4.69 | 0.0481 |
| | Residual | $2.67 \times 10^8$ | 14 | $1.91 \times 10^7$ | | |
| | Aberrant term | $2.10 \times 10^8$ | 8 | $2.62 \times 10^7$ | 2.77 | 0.1159 |
| | The sum | $4.83 \times 10^9$ | 22 | | | |

**Table 6.** Variance analysis of power.

| Test Index | Source | Sum of Squares | Freedom | Mean Square | F Value | p Value |
|---|---|---|---|---|---|---|
| | Model | 39.45 | 8 | 4.93 | 123.23 | <0.0001 |
| | $X_1$ | 2.71 | 1 | 2.71 | 67.84 | <0.0001 |
| | $X_2$ | 2.95 | 1 | 2.95 | 73.79 | <0.0001 |
| | $X_4$ | 20.00 | 1 | 20.00 | 499.82 | <0.0001 |
| Working power | $X_1X_2$ | 12.06 | 1 | 12.06 | 301.46 | <0.0001 |
| | $X_1X_4$ | 1.14 | 1 | 1.14 | 28.58 | 0.0001 |
| | $X_2X_4$ | 0.98 | 1 | 0.976 | 24.39 | 0.0002 |
| | $X_1^2$ | 0.87 | 1 | 0.869 | 21.72 | 0.0004 |
| | $X_4^2$ | 0.51 | 1 | 0.51 | 12.84 | 0.0030 |
| | Residual | 0.56 | 14 | 0.04 | | |
| | Aberrant term | 0.31 | 8 | 0.04 | 0.90 | 0.5693 |
| | The sum | 40.01 | 22 | | | |

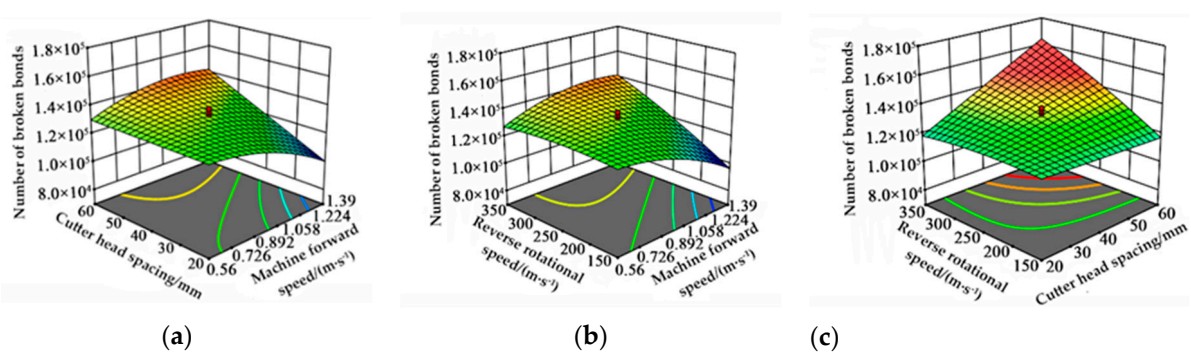

(**a**)       (**b**)       (**c**)

**Figure 11.** Effects of interaction factors on number of broken bonds. (**a**) Interaction between cutter head spacing and forward speed; (**b**) interaction between reverse rotation speed and forward speed; (**c**) interaction between reverse rotation speed and cutter head spacing.

As shown in Figure 11b, when the reverse rotation speed was at the initial stage, the relationship between the forward speed of the implement and the number of broken bonding keys was negative. However, with the increase in the reverse rotation speed, the relationship between the forward speed of the implement and the number of broken bonding keys changed from a negative correlation to a positive correlation. The reason is that the increase in reverse speed increases the frequency of the straw being cut, and the impact of the reverse speed on the number of broken bonds is greater than the forward speed of the machine. When the forward speed of the machine tool is the initial value, the number of broken bonds increases with the increase in the reverse speed, and the number of broken bonds increases more obviously with the increase in the forward speed of the machine tool.

As shown in Figure 11c, when the reverse rotation speed was in the initial stage, the increase in cutter head spacing had little effect on the number of broken bonds; with the increase in reverse rotation speed, the number of bond key fracture increased with the increase in cutter head spacing. When the cutter head spacing was at the initial stage, the number of broken bonds increased with the increase in reverse rotation speed. However, with the increase in cutter head spacing, the number of broken bonds increased with the increase in cutter head spacing.

The objective function was optimized and solved, and the optimal parameter combination was obtained as follows: the forward speed of the tool was 0.98 m·s$^{-1}$, the cutter head spacing was 60 mm, the forward rotation speed was 150 r·min$^{-1}$, and the reverse rotation speed was 313 r·min$^{-1}$. Under this condition, the predicted values were as follows: the number of broken bonding keys was 157,606 and the operating power was 6 kw. According to the optimal parameters, the test results showed that the number of broken bonds was 152,848 and the operating power was 5.86 kw. The relative error with the predicted number of broken bonds was 3.02% and the relative error with the predicted operating power was 2.33%. The verification test showed that the results of the test values were basically consistent with the predicted values, and the optimal parameter combination can be used as the parameter basis for the prototype trial production.

$$\begin{cases} \max Y_1(X_1, X_2, X_3, X_4) \\ Y_2(X_1, X_2, X_3, X_4) \leq 6 \\ \text{s.t} \begin{cases} 0.56 \leq X_1 \leq 1.39 \\ 20 \leq X_2 \leq 60 \\ 150 \leq X_3 \leq 350 \\ 150 \leq X_1 \leq 350 \end{cases} \end{cases} \tag{15}$$

### 3.2. Test Results and Analysis of the Fertilizer Discharge Device

The test results are shown in Figure 12. With the increase in rotating speed, the variation coefficient of the fertilizer discharge uniformity decreased. When the rotating speed reached 70 r/min, the variation coefficients of the ladder arc straight groove, arc straight groove, and arc inclined groove were 0.88%, 1.36%, and 1.46%, respectively. Therefore, the ladder arc straight groove type was selected.

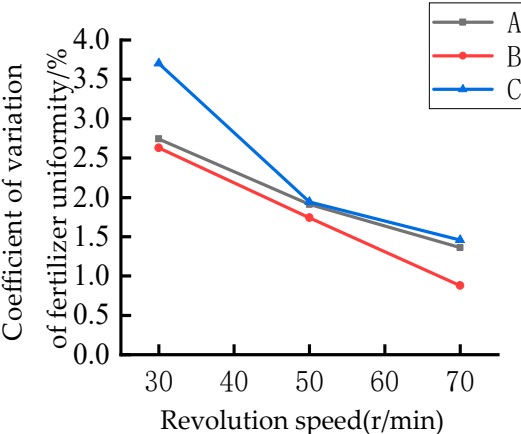

**Figure 12.** Effects of different types of outer-groove wheel on the fertilizer uniformity variation coefficient. Note: in the drawing, A is the circular arc straight groove type, B is the ladder arc straight groove type, and C is the circular arc inclined groove type.

### 3.3. Test Results and Analysis of the Opener Device

The test results are shown in Figure 13 and the furrow shape of the opener is shown in Figure 14. The average resistance of the hoe opener, double disc opener, and core-share opener was 76.3, 271.24, and 104.15 N, respectively. The furrow widths were 21.3, 52, and 61.33 mm. The soil disturbances were 8.45%, 20.63%, and 24.63%. The furrow bottom was flat, uneven, and flat. To sum up, combined with resistance, ditch shape, and ditch width analysis, the hoe ditcher was the best.

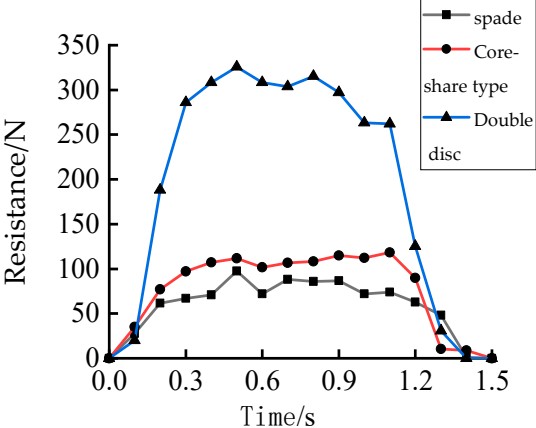

**Figure 13.** Resistance of the opener.

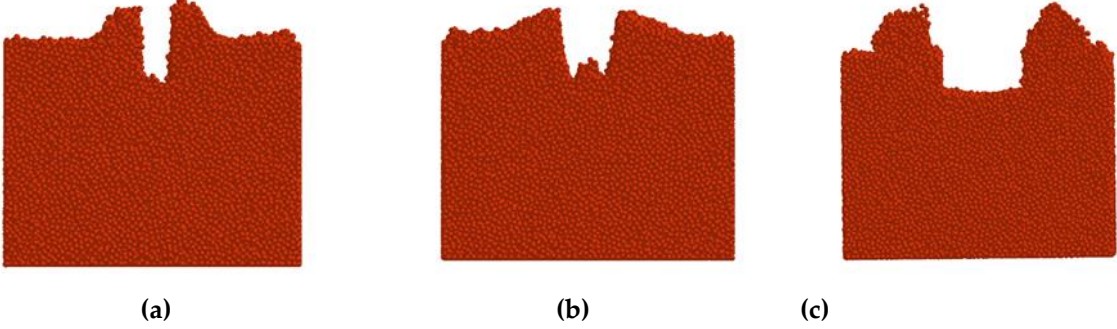

(a)          (b)          (c)

**Figure 14.** Furrow shape. (**a**) Hoe opener; (**b**) double disc opener; (**c**) core-share opener.

*3.4. Field Test*

3.4.1. Test Conditions and Equipment

The experiment was carried out on 5 November 2021 in the conservation tillage experiment field (N24°50′56″, E 102°51′49″) of Kunming University of Science and Technology, Chenggong District, Kunming City, Yunnan Province, with altitude of 1932 m and atmospheric pressure of 80,122 Pa. The soil texture of the test site was red soil. The average soil moisture content in the 0~15 cm soil layer of the test field was 21.58%, the average soil bulk density was 1.21 g·cm$^{-3}$, and the average soil solidity was 1105 kPa. During the experiment, the daily average temperature was 20 °C, there was no rainfall, the maximum wind speed was 2.50 m·s$^{-1}$, the average wind speed was 1.03 m·s$^{-1}$, the relative humidity was 54.4%, the dew point temperature was 10 °C, and wet-bulb temperature was 14 °C. The former crop was corn, the moisture content of straw was 69.73%, the coverage of straw was 86%, and the coverage of straw was 1.63 kg·m$^{-2}$. Corn straw was not crushed, with the whole stalk covering the ground instead.

The test equipment mainly included: an SC900 digital soil firmness meter (0~7000 kPa), a bi-directional rotating stubble no-tillage planter, and a Dongfanghong 504 tractor.

3.4.2. Test Indicators and Methods

According to the national standard GB/t 20865-2017 no (less) tillage fertilizing seeder [36] and the agricultural industry standard NY/t 1003-2006 technical specification for quality evaluation of fertilizing machinery [37], the main test indicators of the field test included the straw cutting rate, trafficability of machinery, seed furrow size, soil disturbance, seed fertilizer spacing qualification rate, consistency coefficient of variation of fertilizer discharge in each row, stability coefficient of variation of total fertilizer discharge, uniformity coefficient of variation of sowing, etc. The machines and tools were tested with the best parameters of the simulation test.

- Straw cutting rate.

Nine measuring areas were selected from the three working journeys, and the length of each measuring area was 1 m. The number of cut straws and the total number of straws in each measuring area were counted, and the straw cutting rate was obtained according to Equation (16).

$$P = \frac{N_D}{N_T} \times 100\% \tag{16}$$

In the formula, P is the straw cutting rate, %; $N_d$ is the amount of straw cut in the measurement area; $N_T$ is the total amount of straw in the measurement area.

- Trafficability of machinery.

Three strokes were operated in the measurement area and the length of each measuring area was 40 m. It was observed during the operation whether the machine could continuously operate normally and the blockage degree of straw to the machine.

- Furrow size and soil disturbance.

After the completion of no-tillage sowing, the furrow size of three working strokes was measured. Three measuring points were randomly selected for each stroke, and nine measuring points were selected. The soil disturbance was calculated according to the Formula (17).

$$\eta = \frac{w_s}{W} \times 100\% \tag{17}$$

In the formula, η is the soil disturbance, %; $w_s$ is the width of furrow surface, cm; W for row spacing, cm.

- Qualified rate of sowing depth and fertilization depth and spacing.

The seeding depth and fertilization depth of three strokes were measured in the completed area, and nine points were measured at three points per stroke. The distance

between the seed and surface is the sowing depth, the distance between the fertilizer and surface is the fertilization depth, and the difference between fertilization depth and sowing depth is seed and fertilizer spacing.

- Coefficient of variation of sowing uniformity.

The ditcher and soil covering device of the no-tillage planter was removed, and the seeding operation was carried out on the ground without straw. A total of six rows were sown in three plots. After the completion of the sowing operation, the three plant spacing was taken as a section (the theoretical plant spacing is 14 cm), and the number of seeds in each section was measured. Ten sections were measured in each plot, and the coefficient of variation of sowing uniformity was calculated according to Equations (18) to (21).

$$D = \sum_{i=1}^{3} D_i \tag{18}$$

$$X = \frac{1}{3} \sum_{i=1}^{3} X_i \tag{19}$$

$$G = \sqrt{\frac{1}{D} \sum (x - X)^2} \tag{20}$$

$$V = \frac{G}{X} \tag{21}$$

In the above formula, D is the total number of segments; $D_i$ is the number of segments per zone; X is the average number of seeds per segment; $X_i$ is the average number of seeds in each section of each region; X is the number of seeds per segment; G is the standard deviation; N is the number of rows; V is the coefficient of variation of seeding uniformity.

- Fertilizing performance.

The no-tillage planter was erected on a flat and hard site, so that the ground wheel left the ground, and the variation coefficient of the consistency of each row of fertilizer discharge and the variation coefficient of the stability of the total fertilizer discharge were measured. The coefficient of variation of the consistency of the fertilizer discharge of each row and the coefficient of variation of the stability of the total fertilizer discharged were calculated according to the Formulas (22)~(25).

$$\eta = \frac{50(1 - \delta)}{\pi D} \tag{22}$$

In the formula, η is the number of rotation rings of the ground wheel; D is wheel diameter, m; δ is slip ratio, %.

$$x = \frac{1}{n}(x_1 + x_2 + \cdots + x_i) \tag{23}$$

$$S = \sqrt{\frac{\sum_{i=1}^{n}(x_i - x)^2}{n - 1}} \tag{24}$$

$$CV = \frac{S}{x} \times 100\% \tag{25}$$

In the formula, $x_i$ is the average or total fertilizer discharge per row, g; x is the average of the average fertilizer discharge per row or the average of the total fertilizer discharge per row, g; n is the number of rows or times measured; S is the standard deviation of the consistency of each row or the stability of the total fertilizer discharged; V is the coefficient of variation for the consistency of fertilizer discharge of each row or the coefficient of variation for the stability of total fertilizer discharged.

### 3.4.3. The Test Results and Analysis

1. First item: straw cutting rate.

The experimental results of the straw cutting rate of the anti-blocking device are shown in Table 7.

**Table 7.** Measurement results of the straw cutting rate.

| Numbering | $P_1$/% | Numbering | $P_1$/% |
|---|---|---|---|
| 1 | 88.24 | 6 | 100.00 |
| 2 | 92.86 | 7 | 91.67 |
| 3 | 92.31 | 8 | 100.00 |
| 4 | 88.89 | 9 | 100.00 |
| 5 | 100.00 | Mean value | 95.72 |

The test results showed that it met the requirements of conservation tillage in Southwest China.

2. Second item: trafficability of machinery.

The operation process is shown in Figure 15. During the three working strokes, the planter did not have severe blockage, and the trafficability of the whole machine was qualified. After the completion of the seeding operation, the surface straw was cut off, the seeding belt was clean, and there were no seeds exposed to the ground.

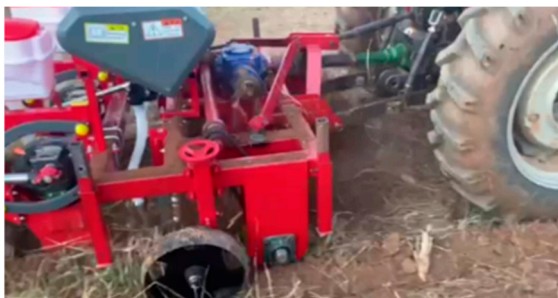

**Figure 15.** Operation process.

3. Third item: furrow size and soil disturbance.

The size of the seed furrow was measured after the operation. The measurement process is shown in Figure 16. The sowing row spacing was 60 cm. The soil disturbance was calculated. The test results are shown in Table 8. According to the requirements of relevant standards, the soil disturbance of the no-tillage planter is ≤ 40%, so the soil disturbance of the bi-directional rotating stubble type no-tillage planter met the requirements of national standards.

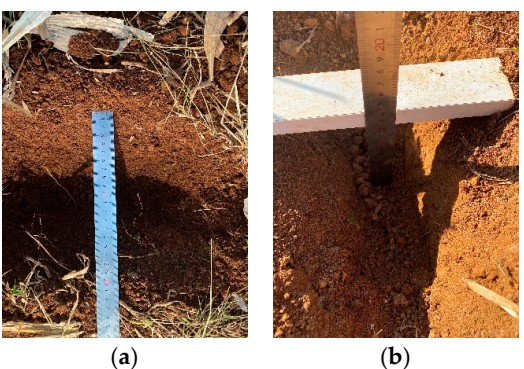

(a)        (b)

**Figure 16.** Measurement of seed furrow size. (**a**) Furrow width; (**b**) furrow depth.

**Table 8.** Measurement results of the seed furrow size.

| Test Indicators | Furrow Depth/cm | Width of Seed Furrow Bottom/cm | Width of Seed Furrow Surface/cm | Soil Disturbance/% |
|---|---|---|---|---|
| Mean value | 10.8 | 6.5 | 19.1 | 31.87 |
| Standard deviation | 1.33 | 1.03 | 2.04 | 0.034 |
| Coefficient of variation | 12.34% | 15.82% | 10.64% | 10.64% |

4.   Fourth item: qualified rate of seeding depth, fertilizer depth, and spacing.

The sowing depth, fertilization depth, and spacing between seeds and fertilizers were measured. The process is shown in Figure 17, and the results are shown in Table 9. The theoretical sowing depth was 5 cm and the theoretical fertilization depth was 10 cm. According to Table 9, the qualified rates of sowing depth, fertilization depth, and seed fertilizer spacing were 88.89%, 100%, and 100%, respectively, which all met the relevant standards.

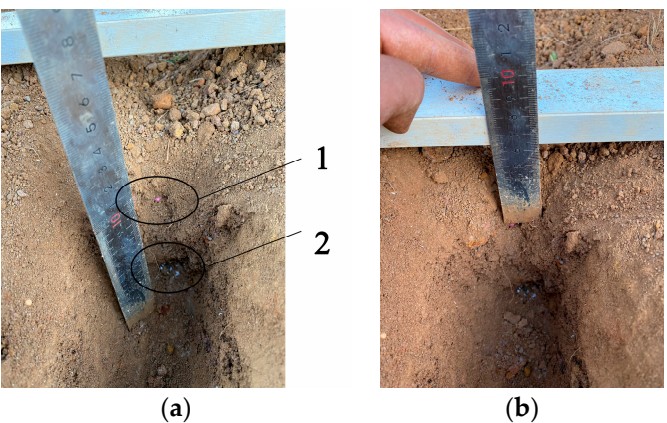

(**a**)                    (**b**)

**Figure 17.** Measurement of seeding depth and fertilization depth. (**a**) Seed depth measurement; (**b**) fertilizer depth measurement; 1. Seed; 2. Fertilizer.

**Table 9.** Measurement results of seed depth, fertilizer depth, and seed–fertilizer spacing.

| Test Indicators | Seeding Depth/cm | Fertilization Depth/cm | Spacing between Seeding and Fertilization/cm |
|---|---|---|---|
| Average | 5.4 | 10.1 | 4.7 |
| Standard deviation | 0.59 | 0.85 | 0.42 |
| Coefficient of variation | 10.96% | 8.50% | 9.07% |
| Qualified rate | 88.89% | 100% | 100% |

5.   Fifth item: coefficient of variation of sowing uniformity.

The coefficient of variation of seeding uniformity of the bi-directional rotating stubble no-tillage planter is shown in Table 10. According to the requirements of relevant standards, the coefficient of variation of sowing uniformity should be ≤45%, so the coefficient of variation of sowing uniformity of the bi-directional rotating stubble no-tillage planter met the requirements of national standards.

**Table 10.** Measurement result of the variation coefficient of seeding uniformity.

| Test Index | Number of Seeds in Zone 1 | Number of Seeds in Zone 2 | Number of Seeds in Zone 3 |
|---|---|---|---|
| Average | 3 | 2.8 | 3.4 |
| Total average number of grains | | 3.07 | |
| Total standard deviation | | 0.68 | |
| Coefficient of variation of seeding uniformity | | 22.15% | |

6.  Sixth item: fertilizing performance.

The fertilizer discharge performance of the bi-directional rotating no-tillage planter was measured, and the results are shown in Table 11. According to relevant regulations, the coefficient of variation of consistency of fertilizer discharge of each row was $\leq 13.0\%$, and the coefficient of variation of stability of total fertilizer discharge was $\leq 7.8\%$. It can be seen from Table 11 that the coefficient of variation of consistency of fertilizer discharge of each row and the stability of total fertilizer discharge is qualified.

**Table 11.** Test results of the variation coefficient of seeding uniformity.

| Test Index | Amount of Fertilizer in the First Row/g | Amount of Fertilizer in the Second Row/g | Total Fertilizer Discharged/g |
|---|---|---|---|
| Average amount of fertilizer | 4778.9 | 5076.8 | 9855.6 |
| Coefficient of variation for consistency of amount of fertilizer per row | | 4.27% | |
| Variation coefficient of stability of total amount of fertilizer | | 1.52% | |

## 4. Conclusions

Due to the large amount of straw coverage and the problem of whole straw coverage in Southwest China, the popularization and application of no-tillage planters in Southwest China are limited. Solving the problem of no-tillage sowing in Southwest China is of great significance for improving economic benefits and production efficiency. According to the bionic principle, the image of mouthparts of B. horsfieldi was obtained through a microscope, and the edge curve coordinate points were obtained by using MATLAB to process the image. The coordinate points were imported into Origin to obtain the curve equation, and the curves and edges of left and right disc cutter were designed. The discrete element model was established, and the quadratic regression rotation orthogonal combination test with four factors and five levels was carried out to obtain the optimal parameters of the bi-directional rotating stubble-cutting and anti-blocking device. Comparative tests were carried out to compare the fertilizer discharge performance of three types of fertilizer apparatus, namely, circular arc straight groove type, ladder arc straight groove type, and circular arc inclined groove type, and the trenching performance of three types of openers were compared, namely, hoe type, double disc type, and core share type. The prototype was trial manufactured with the best parameters for field tests.

1. The curve equation of the mouthpart contour of B. horsfieldi was obtained with the bionic principle. The bi-directional rotating stubble cutting anti-blocking device was designed, and the curves of left and right disc cutters were designed.

2. A discrete element model was established to optimize the key components of anti-blocking device, fertilizer apparatus, and opener. The machine forward speed of $0.98 \text{ m·s}^{-1}$, the cutter spacing of 60 mm, the forward rotation speed of $150 \text{ r·min}^{-1}$, and the reverse rotation speed of $313 \text{ r·min}^{-1}$ were the optimal parameters. The coefficient of variation of fertilizer uniformity of the ladder arc straight groove type outer groove wheel fertilizer distributor is the smallest, and the ditching resistance and soil disturbance of the hoe opener are the smallest.

3. The results of field experiment showed that when the straw mulching amount was 1.63 kg·m$^{-2}$, the average straw cutting rate was 95.72%, the passing ability of machine was good, the soil disturbance amount was 31.87%, the qualified rate of sowing depth was 88.89%, the qualified rate of fertilization depth and spacing was 100%, the variation coefficient of sowing uniformity was 22.15%, and the variation coefficient of total fertilizer discharge stability was 1.52%.

The designed bi-directional rotating stubble-cutting no-tillage planter meets the requirements of no-tillage sowing operation in Southwest China and can solve the problems of the large amount of straw coverage and whole straw coverage in Southwest China and improve agricultural production efficiency. This research can provide a reference for the design and improvement of no-tillage planters under the conditions of a large amount of straw coverage and whole straw coverage. The terrain in Southwest China is mostly hilly and mountainous. In order to meet the requirements of no-tillage sowing operation on different slopes, it is necessary to further study the profiling mechanism of no-tillage planters.

**Author Contributions:** Conceptualization, H.Z. (Huibin Zhu) and X.W.; methodology, H.Z. (Huibin Zhu); software, H.Z. (Huibin Zhu); validation, C.Q., S.M., H.Z. (Haoran Zhao), X.Z. and H.L.; resources, L.B.; data curation, H.Z. (Huibin Zhu); writing—original draft preparation, H.Z. (Huibin Zhu); writing—review and editing, H.Z. (Huibin Zhu) and X.W.; project administration, L.B. All authors have read and agreed to the published version of the manuscript.

**Funding:** This research was funded by the National Natural Science Foundation of China, grant number 51865022.

**Institutional Review Board Statement:** Not applicable.

**Informed Consent Statement:** Not applicable.

**Data Availability Statement:** The analyzed datasets are available from the corresponding author on reasonable request.

**Conflicts of Interest:** The authors declare no conflict of interest.

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
