# Peer review of "Design and Experimental Study of a Bi-Directional Rotating Stubble-Cutting No-Tillage Planter"

_agriculture, doi:10.3390/agriculture12101637_

Round 1

Reviewer 1 Report

Authors should describe more clearly the relevance of the research work, what is the scientific contribution?

Authors should revise the wording and English of the manuscript and improve it.

Authors should check current reference

Authors should be more specific in their conclusions, highlighting the achievements that contribute to the work and the contributions generated.

Author Response

Point 1: Authors should describe more clearly the relevance of the research work, what is the scientific contribution?

Response 1: According to expert’s opinions, relevant work has been described again in the introduction of the manuscript, and scientific contributions have been added in the conclusion?

Point 2: Authors should revise the wording and English of the manuscript and improve it.

Response 2: According to expert’s opinions, the wording and English of the manuscript have been revised.

Point 3: Authors should check current reference

Response 3: According to expert’s opinions, the references are rechecked in the text.

Point 4: Authors should be more specific in their conclusions, highlighting the achievements that contribute to the work and the contributions generated.

Response 4: According to expert’s opinions, the conclusions have been redescribed, and the achievements that contribute to the work and the contributions generated have been added.

Reviewer 2 Report

- please improve the introduction;

- please improve the conclusions;

- please expand the explanations for fig 8 (more details please);

- please check the english language.

Author Response

Point 1: please improve the introduction;

Response 1: According to expert’s opinions, the introduction has been improved.

Point 2: please improve the conclusions;

Response 2: According to expert’s opinions, the conclusion has been improved.

Point 3: please expand the explanations for fig 8 (more details please);

Response 3: According to expert’s opinions, Figure 8 has been expanded and explained (now Figure 11).

Point 4: please check the english language.

Response 4: According to expert’s opinion, English language has been rechecked.

Reviewer 3 Report

In this study, optimal cutting curve was designed and the equation fitting of the contour curve of the B. horsfiel Hope the mouthpiece was carried out through the bionic principle based on the principle of supported cutting. Manuscript needs improvement. My suggestions are follows:

·         Similar studies have not been examined before. Authors should emphasize the originality of the study with reference to the literature.

·         Reference should be made to the literature for theoretical explanations in Section 2.2.1.

·         References 5-9 are not cited in the text.

·         Suggestions for future work are expected in the "Conclusion" section.

·         No space is left between the figure and table captions and the text.

·         Figure references should be given as Figure, not Fig.

Author Response

Point 1: Similar studies have not been examined before. Authors should emphasize the originality of the study with reference to the literature.

Response 1: According to expert’s opinions, references have been cited in the bionic part.

Point 2: Reference should be made to the literature for theoretical explanations in Section 2.2.1.

Response 2: According to expert’s opinions, the theoretical explanation in Section 2.2.1 has been made, and references have been cited.

Point 3: References 5-9 are not cited in the text.

Response 3: According to expert’s opinions, references 5-9 have been cited and the serial numbers are changed to 12-14, 17-18.

Point 4: Suggestions for future work are expected in the "Conclusion" section.

Response 4: According to the expert’s opinions, we make suggestions for the future work at the conclusion part. “In order to meet the requirements of no-tillage sowing operation on different slopes, it is necessary to further study the profiling mechanism of no-tillage planter.”

Point 5: No space is left between the figure and table captions and the text.

Response 5: According to expert’s opinion, the spaces have been added between the figure and table captions and the text.

Point 6: Figure references should be given as Figure, not Fig.

Response 6: According to expert’s opinions, the figure reference has been changed to be given as figure.

Reviewer 4 Report

The reviewer did not fully understand the content of the paper due to the spelling and words used in the paper. Only the following questions and suggestions are put forward.

1. It is suggested to change the name of the planter to bi-directional rotating stubble-cutting no-tillage planter.

2. Please introduce the method of obtaining bionic curve.

3. Please correct the header ' Sun of squares ' in Table 6 and the headlines ' Test results and analysis of fertilizer discharge device ' in Sections 3.2 and 3.3.

4. Why the soil disturbance coefficient in formula 17 does not use the algorithm recommended by national standard of PRC (GB/T 24675.2-2009).

5. Why the difference between the average values of Ditch depth and Seeding depth in Table 8 and Table 9 is so large, please explain the measurement experiment scheme in detail.

Author Response

Point 1: It is suggested to change the name of the planter to bi-directional rotating stubble-cutting no-tillage planter.

Response 1: According to expert’s opinions, the name of the planter has been changed to“bi-directional rotating stubble-cutting no-tillage planter.”

Point 2: Please introduce the method of obtaining bionic curve.

Response 2: According to expert’s opinions, the method of obtaining bionic curve has been introduced in 2.2.1.

Point 3: Please correct the header ' Sun of squares ' in Table 6 and the headlines ' Test results and analysis of fertilizer discharge device ' in Sections 3.2 and 3.3.

Response 3: According to expert’s opinions, the titles of Table 6, "Sum of Squares", 3.2, "Test results and analysis of fertiliser discharge device" and 3.3, " Test results and analysis of opener device" have been changed in the text.

Point 4: Why the soil disturbance coefficient in formula 17 does not use the algorithm recommended by national standard of PRC (GB/T 24675.2-2009).

Response 4: The national standard of the People's Republic of China (GB/T 24675.2-2009) is the standard for soil disturbance of subsoiling machinery. This paper is about no-tillage planter, so it is necessary to correspond to the national standard of no tillage planter, GB/T 20865-2017 No (Low) Tillage Fertilization Planter

Point 5: Why the difference between the average values of Ditch depth and Seeding depth in Table 8 and Table 9 is so large, please explain the measurement experiment scheme in detail.

Response 5: In fact, the before titles of Table 8 and Table 9 are wrongly written. The titles of Table 8 should be " Furrow depth/cm、Width of seed furrow bottom /cm、Width of seed furrow surface/cm、Soil disturbance/%". The titles of Table 9 should be "Seeding depth/cm、Fertilization depth/cm、Spacing between seeding and fertilization/cm ". The experimental measurement method has been described in 3.4.2.

Round 2

Reviewer 3 Report

Now, the manuscript appears ready for publication. It is acceptable as it is.

Reviewer 4 Report

no more comments and suggestions